

# Identification and characterization of novel sesquiterpene synthases TPS9 and TPS12 from *Aquilaria sinensis*

Cuicui Yu[1], Shixi Gao[1], Mei Rong[1], Mengjun Xiao[1], Yanhong Xu[1] and Jianhe Wei[1,2]

[1] Key Laboratory of Bioactive Substances and Resources Utilization of Chinese Herbal Medicine, Ministry of Education & National Engineering Laboratory for Breeding of Endangered Medicinal Materials, Institute of Medicinal Plant Development, Chinese Academy, Beijing, China
[2] Hainan Provincial Key Laboratory of Resources Conservation and Development of Southern Medicine & Key Laboratory of State Administration of Traditional Chinese Medicine for Agarwood Sustainable Utilization, Hainan Branch of the Institute of Medicinal Plan, Hainan, China

## ABSTRACT

Sesquiterpenes are characteristic components and important quality criterions for agarwood. Although sesquiterpenes are well-known to be biosynthesized by sesquiterpene synthases (TPSs), to date, only a few TPS genes involved in agarwood formation have been reported. Here, two new TPS genes, namely, *TPS9* and *TPS12*, were isolated from *Aquilaria sinensis* (Lour.) Gilg, and their functions were examined in *Escherichia coli* BL21(DE3), with farnesyl pyrophosphate (FPP) and geranyl pyrophosphate (GPP) as the substrate of the corresponding enzyme activities. They were both identified as a multiproduct enzymes. After incubation with FPP, TPS9 liberated $\beta$-farnesene and cis-sesquisabinene hydrate as main products, with cedrol and another unidentified sesquiterpene as minor products. TPS12 catalyzes the formation of $\beta$-farnesene, nerolidol, $\gamma$-eudesmol, and hinesol. After incubation with GPP, TPS9 generated citronellol and geraniol as main products, with seven minor products. TPS12 converted GPP into four monoterpenes, with citral as the main product, and three minor products. Both *TPS9* and *TPS12* showed much higher expression in the two major tissues emitting floral volatiles: flowers and agarwood. Further, RT-PCR analysis showed *TPS9* and *TPS12* are typical genes mainly expressed during later stages of stress response, which is better known than that of chromone derivatives. This study will advance our understanding of agarwood formation and provide a solid theoretical foundation for clarifying its mechanism in *A. sinensis*.

Corresponding authors
Yanhong Xu, xuyanhong99@163.com
Jianhe Wei, wjianh@263.net

## INTRODUCTION

Agarwood is a dark resinous substance formed in the branches and trunks of *Aquilaria* and *Gyrinops* species of the Thymelaeaceae family upon stimulation of the tree by various stress factors, including mechanical wounding, microbial infection, cutting, lightning strikes, and chemical wounding (*Mei et al., 2013*). Agarwood is popular in many countries

because of its high commercial value as an incense, as a scent, as an ornamental wood, and as a traditional medicine (*Cui et al., 2013*). However, agarwood formation occurs slowly under natural conditions; indeed, it may take even decades. Thus, because of its immense value and rarity, high-quality agarwood is in short supply and may be up to US $100,000 per kilogram in the international market (*Persoon & Van Beek, 2008*). Consequently, to protect plant genetic resources, nine *Aquilaria* species have been included in the IUCN Red List (*Wyn & Anak, 2010*). In particular, *A. sinensis* (Lour.) Gilg. is the main plant resource for producing agarwood in China, with the species mainly distributed in South China (*Qi, Lin & Hu, 2000*). To induce qualified agarwood formation, our lab has developed a simpler and efficient method called the whole-tree agarwood-inducing technique (Agar-Wit) (*Liu et al., 2013*).

Previous phytochemical research showed that sesquiterpenoids and 2-(2-phenylethyl) chromones are the main active constituents in agarwood (*Chen et al., 2011b*). Further-more, sesquiterpenoid content has been identified as an important criterion defining agarwood quality (*Chen et al., 2012*; *Li et al., 2016*; *Wang et al., 2016*). Therefore, unveiling the biosynthesis and regulation of sesquiterpenes and chromones in *A. sinensis* is of paramount importance for identifying the mechanism of agarwood formation.

The biosynthesis of sesquiterpene is better known than that of chromone derivatives. Further, sesquiterpene can reportedly be synthesised *via* two different routes, namely, either through the mevalonic acid (MVA) pathway (*Gardner & Hampton, 1999*; *McCaskill & Croteau, 1995*), or through the methylerythritol phosphate (MEP) pathway (*Lichten-thaler, 1999*). Terpene synthases (TPSs) are able to catalyse the formation of C5, C10, C15 and C20 terpene skeletons from allylic prenyl diphosphate intermediates in the terpene biosynthesis pathway, such as geranyl diphosphate (GPP, C10), farnesyl diphosphate (FPP, C15) and geranylgeranyl diphosphate (GGPP, C20) (*Davis & Croteau, 2000*; *Takahashi & Koyama, 2006*). Additionally, the varies types of terpenoids is mainly due to the great diversity of the terpene synthase gene family (*Chen et al., 2011a*; *Rohmer, 1999*). Various types of sesquiterpene synthases (TPS) have been cloned from monocots, such as *Artemisia annua* L. (*Kanagarajan et al., 2012*), and maize (*Schnee et al., 2002*); dicots, such as valerian (*Kwon et al., 2014*), grape (*Dueholm et al., 2019*), ginseng (*Khorolragchaa et al., 2010*), lavender (*Jullien et al., 2014*), snapdragon (*Nagegowda et al., 2008*), castor bean (*Xie, Kirby & Keasling, 2012*), and sandalwood (*Srivastava et al., 2015*); and fungi like *termitomyces* (*Burkhardt et al., 2019*). The same sesquiterpene synthase produces various intermediates through controlled catalysis; for example, *Steele et al. (1998)* reported 52 different sesquiterpenes catalyzed by $\gamma$-humulene synthase cloned from *Abies grandis*.

To date, 180 sesquiterpenes have been detected in *Aquilaria* plants (*Ahmaed & Kulkarni, 2017*). However, only a few sesquiterpene synthase genes involved in sesquiterpene formation in *Aquilaria* plants have been identified and cloned, including *AcC2*, *AcC3*, *AcC4*, *AcL154* (*Kumeta & Ito, 2010*), *GS2*, *GS3*, *GS4* (*Kurosaki et al., 2015*), *ASS1*, *ASS2*, *ASS3* (*Xu et al., 2013*), *As-SesTPS* (*Ye et al., 2018*). The main product of catalysis by these enzymes was identified as $\delta$-guaiene, which is a type of sesquiterpene.

Therefore, as the identification of new genes is of great significance for analysing the formation and diversity of agarwood sesquiterpenes in *A. sinensis*, two novel TPS genes,

*TPS9* and *TPS12*, were cloned from *A. sinensis*, and their *in vitro* catalytic function, as well as their expression characteristics, were studied after various stress treatments, including methyl jasmonate (MeJA), heat stress, salicylic acid (SA), ABA (abscisic acid), mannitol, $H_2O_2$ and NaCl, to further investigate the mechanism of agarwood formation. *TPS9* and *TPS12* showed very low identity with heterologously produced sesquiterpene synthases, and their expression profiles after Agar-Wit treatment and catalysis were very different from *ASS1*, a sesquiterpene synthase previously identified (*Xu et al., 2021*).

## MATERIALS & METHODS

### Plant materials and treatment

*A. sinensis* trees were grown in the Hainan branch of the Institute of Medicinal Plant Development (IMPLAD). Different tissues were collected from two-year-old trees, and materials for transcription sequencing were obtained from seven-year-old trees treated using the whole-tree agarwood-inducing technique (Agar-Wit) (*Wei et al., 2010*). All collected materials were stored in liquid nitrogen until further analysis. *A. sinensis* calli were induced from fresh young leaves as previously described by *Liu et al. (2015a)*. Healthy, faint yellow, and compact calli were sub-cultured in darkness at 25 °C in Murashige-Skoog (MS) medium. For heat treatment, the same well-grown calli were transferred from 25 °C to 42 °C. For other stress treatments, calli were transferred to MS medium supplemented with 100 μM MeJA, 100 μM SA, 100 μM ABA, 400 mM mannitol, 50 mM $H_2O_2$, 300 mM NaCl, and incubated for different time points. Untreated calli were simultaneously sampled over the same period and used as controls. All samples were shock-frozen in liquid nitrogen and stored at −80 °C for qRT-PCR analysis.

### Bioinformatic analysis and characterization of *TPS9* and *TPS12*

The Expasy Proteomics Server (http://www.expasy.org) was used to calculate physical and chemical parameters of *TPS9* and *TPS12*, such as molecular weight (MW), theoretical isoelectric point (pI), stability, and instability index. Conserved motifs of TPS9 and TPS12 were predicted using Multiple Expectation Maximisation for Motif Elicitation (MEME) with the default parameters. The SWISS-MODEL was used to analyse the molecular modelling of TPS9 and TPS12 proteins. Trans-membrane domains were predicted using TMHMM 2.0 and TMpred software. DNAMAN was used for multiple alignment analysis of the TPS9 and TPS12 amino acid sequences, ExPASy and three softwares, including SCLpredT, Predotar, and Wolf Psort, were used to predict the physiological and biochemical properties and subcellular localisation of TPS9 and TPS12. Phylogenetic analysis was performed using MEGA5 using the neighbour-joining method with 1,000 bootstrap replicates.

### Isolation of total RNA and real-time qRT-PCR

Total RNA was extracted from the treated calli and plants using the Total RNA Rapid Extraction kit RN38-EASYspin Plus (Aidlab, Gdansk, Poland). A total RNA of 1 μg was reverse-transcribed to cDNA using the PrimeScript™ RT Reagent Kit (Takara, Shiga, Japan) according to the protocol described by the manufacturer. The cDNA was stored at
20 °C for qRT–PCR analysis and gene cloning. PCR amplifications were performed using SYBR® Premix Ex Taq™ II (Takara, Shiga, Japan)) on a Light Cycler® 480II (Roche Diagnostics, Indianapolis, IN, USA). The PCR cycling conditions were the same as has described by *Yu et al. (2021)* previously. The glyceraldehyde-3-phosphate dehydrogenase (GADPH) gene was used as an internal control. Three independent biological replicates were included, and the relative expression levels of *TPS9* and *TPS12* were calculated using the $2^{-\Delta\Delta CT}$ method (*Livak & Schmittgen, 2001*).

## Construction of pET21a-TPS9 and pET28a-TPS12

Full-length sequences of *TPS9* and *TPS12* with restriction enzyme sites were amplified using primers F1 (TCTACACCAGCACTTGCCCTCTAC) and R1 (TACAACTC-CTTCACTG CTTCCTGC), F2 (GCATTTCGCTGCTGTTTC), and R2 (AATG-GATTTGAGGTGGGTC). The resultant fragments of *TPS9* and *TPS12* were digested with restriction enzymes XhoI and BamHI and inserted into the expression vectors pET21a and pET28a, respectively, which were also digested with the same restriction enzymes. The ligation product was transformed into *E. coli* DH5 $\alpha$ competent cells and spread on Luria-Bertani (LB) medium containing 100 μg/mL ampicillin and 50 μg/mL kanamycin. Positive clones were extracted and confirmed using digestion and sequencing. All bacterial strains were stored in the −80 °C refrigerator in the laboratory.

## Expression of *TPS9* and *TPS12* in *E. coli*

Recombinant plasmids pET21a-TPS9 and pET28a-TPS12 were transformed into *E. coli* BL21(DE3) competent cells. Empty vectors pET-2a and pET-28a were transformed into *E. coli* BL21(DE3) cells as negative controls. Positive clones were selected and inoculated into LB medium containing 100 μg/ml ampicillin or 50 μg/ml kanamycin. The expression of pET21a-TPS9 and pET28a-TPS12 was induced under different IPTG concentrations (0.2 or 0.5 mM), incubation times (4, 6, 8, or 24 h), and temperatures (16 or 36 °C) to determine the optimal induction conditions. Then, positive clones of pET21a-TPS9 and pET28a-TPS12 were inoculated into LB medium containing 100 μg/mL ampicillin or 50 μg/mL kanamycin on a large scale after $OD_{600}$ reached 0.6−0.8. Bacterial cells were induced with 0.5 mM isopropyl $\beta$-D-thiogalactopyranoside (IPTG) for 6 h at 36 °C and proteins were obtained by ultrasonic decomposition for 1 h. The solution was centrifuged for 20 min at 5,000 rpm and 4 °C, and the supernatant was discarded. The target protein was detected using 10 μL of the sample for separation using 10% SDS-PAGE.

## Western blot analysis

Proteins were separated by 10% SDS-PAGE and transferred onto 0.45 μm polyvinylidene fluoride membranes (Millipore, Burlington, MA, USA). The membranes were blocked with TBST buffer (20 mM Tris–HCl, 150 mM NaCl, and 0.05% (v/v) Tween 20) containing 5% fat-free milk powder and incubated at 4 °C for 4 h. The membrane containing the two expression vectors, TPS9-pET21a and TPS12-pET28a, with His-tag, was incubated overnight at 4 °C with anti-His antibody diluted at 1:2,000 (Transgen, Beijing, China), washed twice with TBST at 4 °C, and incubated with a secondary antibody conjugated to alkaline phosphatase diluted at 1:8,000 (Transgen, Beijing, China).

## Heterologous protein production and enzymatic assays

*E. coli* cells with positive clones of pET21a-TPS9 or pET28a-TPS12 were inoculated into 200 ml LB medium until $OD_{600}$ reached 0.6−0.8. To increase the protein expression level, a longer incubation time was needed, and thus, bacterial cells were induced with 0.5 mM isopropyl IPTG for 12 h at 37 °C. The bacterial solution was concentrated to 10 ml by centrifuging at 4 °C, followed by ultrasonic decomposition for 1 h. The solution was placed into 20 ml sample bottles with 60 μM FPP (Sigma, St. Louis, MO, USA) or 60 μM GPP (Sigma, St. Louis, MO, USA) and kept for 60 min in a water bath at 30 °C. According to the protocol by the manufacturer, vapour from the bottle was extracted with a solid phase micro extraction (SPME) fibre (100 m polydimethylsiloxane, Supelco, Bellefonte, PA, USA) for 90 min in a water bath at 75 °C, following injection of the sample into the gas chromatograph.

## GC-MS analysis

GC-MS analysis was performed using a Perkin Elmer Clarus 600 (Waltham, MA, USA) gas chromatograph equipped with an Agilent DB-5MS capillary column (internal diameter, 30 m × 0.25 mm; film thickness, 0.25 mm) and a Varian 600 mass spectrometer with an ion-trap detector in full scan mode under electron impact ionisation (70 eV), as described by *Liu et al. (2015b)*. The carrier gas was helium and the flow rate was at one mL min $^{-1}$. Samples were injected in the splitless mode at 250 °C. Thehe temperature operating conditions were as follows: 30 °C for 3 min, then increasing to 200 °C at a rate of 4 °C min $^{-1}$, and maintaining that temperature for 5 min (*Chen et al., 2011b*; *Xu et al., 2013*). 1 μL $C_7$-$C_{40}$ n-alkanes was injected separately and ran in the same program as the essential oils.

## Identification of components

Sesquiterpenes were identified by comparing their Kovats retention indices (RI) with those from the literature (*Cavalli et al., 2003*; *Gkinis et al., 2003*; *Siani et al., 2004*; *Tzakou et al., 2004*; *Ghasemi et al., 2005*; *Li et al., 2006*; *Tepe et al., 2006*; *Vujisic et al., 2006*; *Pérez, Navarro & de Lorenzo, 2007*; *Saroglou et al., 2007*; *Basta et al., 2007*; *Hammami, Kamoun & Rebai, 2011*), and comparing the mass spectra obtained with those stored in the NIST MS database and with mass spectra from the literature (*Chen et al., 2011b*; *Liu et al., 2015b*; *Xu et al., 2013*). The Kovats retention indices were determined in relation to a homologous series of n-alkanes ($C_7$–$C_{40}$) under the same operating conditions. Kovats retention indices was calculated as $RI = 100n + 100n\frac{t_x-t_n}{t_{n+1}-t_n}$, where $t_x$ is the retention time of the compound to be measured, $t_n$ is the retention time of the alkane that elutes prior to the compound, $t_{n+1}$ is the retention time of the next eluting alkane, and $n$ is the carbon number for the preceding alkane. Further identification was made by comparing their mass spectra with these stored in NIST 11 and with mass spectra from the literature (*Chen et al., 2011b*; *Liu et al., 2015b*; *Xu et al., 2013*). We did not use an internal standard because it is difficult to choose one that is suitable for all samples.
## RESULTS

### Isolation and bioinformatics analysis of *TPS9* and *TPS12*

Agarwood formation and accumulation under natural conditions may take even decades. In our previous studies, we mainly focused on sesquiterpene synthases induced soon after wounding treatment (*Xu et al., 2014*; *Xu et al., 2021*). However, sesquiterpene synthases induced subsequently, which are key regulators contributing to agarwood accumulation, have not yet been identified. *TPS9* and *TPS12* attracted our attention because transcriptomic studies of *A. sinensis* showed high expression levels of these synthases at the later stages of the plant response to external stimuli, suggesting a potential biological function in agarwood accumulation. Based on the sequences obtained from genome sequencing, we used PCR technique with specific primers to clone two sesquiterpene synthase genes, named *TPS9* and *TPS912*. The isolated cDNAs have open reading frames (ORFs) of 1,383 and 1,632 base pairs, encoding predicted proteins of 460 and 543 amino acid residues, respectively. Aspartate-rich motifs, including the Rx8W, DDxxD, and NSE/DTE motifs, which are conserved in TPSs, were also found to be present in the amino acid sequences of TPS9 and TPS12 (Fig. 1). Predicted molecular weights were approximately 53.3 and 62.9 kDa for TPS9 and TPS12, respectively. Other physiological and biochemical properties of the two proteins are listed in Table S1; further, a three-dimensional structural model constructed using SWISS-MODEL showed that the two structures were similar (Fig. S1). The MEME motif search tool was used to predict the conserved motifs of TPS9 and TPS12; as a result, four putative motifs were identified (Fig. S2). The arginine-rich N-terminal RR(x8)W motif required for cyclisation in TPSs, the highly conserved aspartate-rich DDxxD motif required for $Mg^{2+}$ or $Mn^{2+}$ binding, and the NSE/DTE motifs were found in the amino acid sequences of both TPS9 and TPS12. Analysis using SignalP, TMHMM, and ProtScale predicted that both TPS9 and TPS12 have no transmembrane or signal peptide and are soluble proteins.

   To identify the phylogenetic relationship between TPS9 and TPS12 and subfamilies of the plant TPS family, an unrooted phylogenetic tree of TPS9 and TPS12 in *A. sinensis* and TPSs in 23 other plant species was constructed using MEGAX software with the neighbour-joining method (Fig. 2). Based on amino acid sequence homology, the plant-TPS family is divided into six subfamilies designated TPSa to TPSf (*Bohlmann, Meyer-Gauen & Croteau, 1998*). Our data showed that TPS9 and TPS12 belong to the TPSa family.

### Expression of TPS9 and TPS12 in *E. coli*

To obtain TPS9 and TPS12, full-length *TPS9* and *TPS12* were each ligated into the pET-21a expression vector and coded proteins were expressed in *E. coli* BL21 (DE3). The results showed that *TPS12* did not express any protein. We then changed pET-21a to the pET-28a expression vector, and the *TPS12*-encoded protein was expressed in *E. coli* BL21 (DE3). SDS-PAGE analysis of crude extracts from transformed *E. coli* BL21 (DE3) showed the expected molecular mass with excessive amounts of polypeptides. As shown in Fig. 3, at 0.5 mM IPTG and 36 ° C for 2 h, TPS9 and TPS12 showed expression levels that did not change markedly when incubation periods were extended. After lysis, the

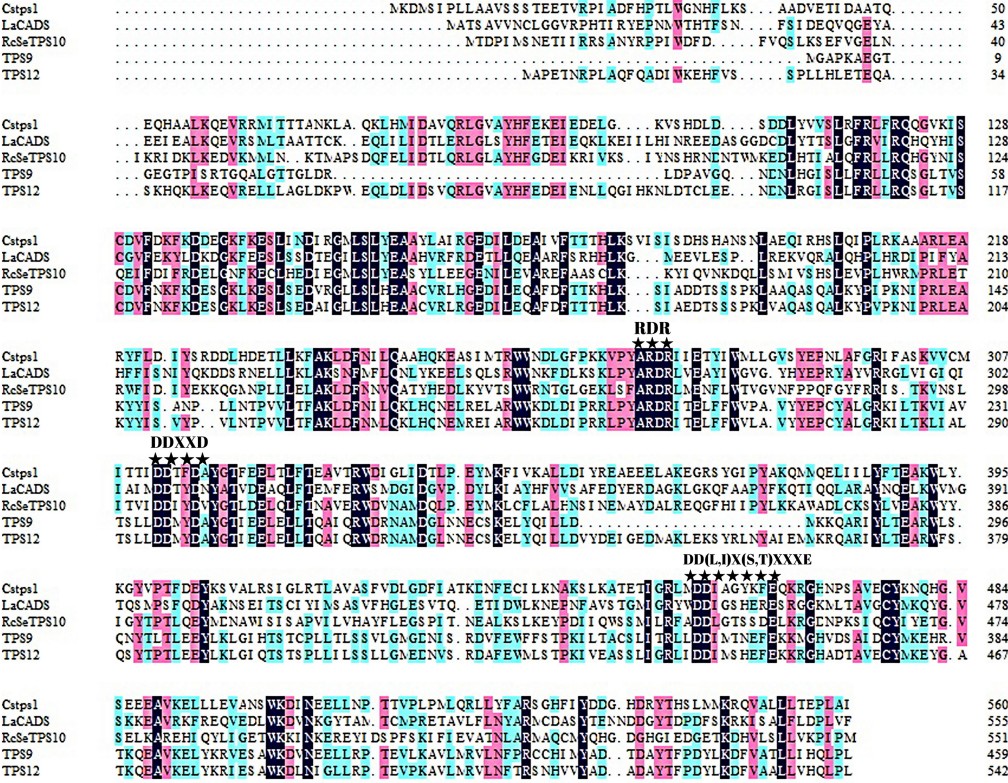

**Figure 1** **Multiple alignment of deduced amino acid sequences of TPS9 and TPS12.** Alignments were performed with DNAMAN. Residues conserved in TPS9 and TPS12 protein are shown with a black background, residues are different in TPS9 and TPS12 protein are shown in blue. The TPS conserved motifs Rx8W, DDxxD, and NSE/DTE ((N/D) Dxx (S/T) xxxE)) are indicated by stars. The numbers on the right indicate the amino acid position.

sediment and supernatant were analysed by 10% SDS-PAGE. The results confirmed that most recombinant pET21a-TPS9 and pET28a-TPS12 proteins were in the non-soluble fractions, contradicting the results predicted by SignalP, TMHMM, and ProtScale. We speculated that the prokaryotic expression system might not be suitable for expressing TPS9 and TPS12, as *TPS9* and *TPS12* are eukaryotic genes.

## Function characterization of TPS9 and TPS12 by *in-vitro* enzyme assay

Analysis by SDS-PAGE showed that most recombinant pET21a-TPS9 and pET28a-TPS12 were inclusion body proteins with no activity. Additionally, Western blot analysis showed a small amount of protein present in the supernatant. As functional enzymes are able to show catalytic activity even when present in small quantities, we attempted to detect such catalytic function *in vivo* using total protein.

As expected, the total protein extract harbouring TPS9 and TPS12 showed activity when FPP was used as a substrate (Table 1 and Fig. 4). The volatile crude extract from *E. coli* BL21 (DE3) was examined by GC-MS, which indicated that the two enzymes are

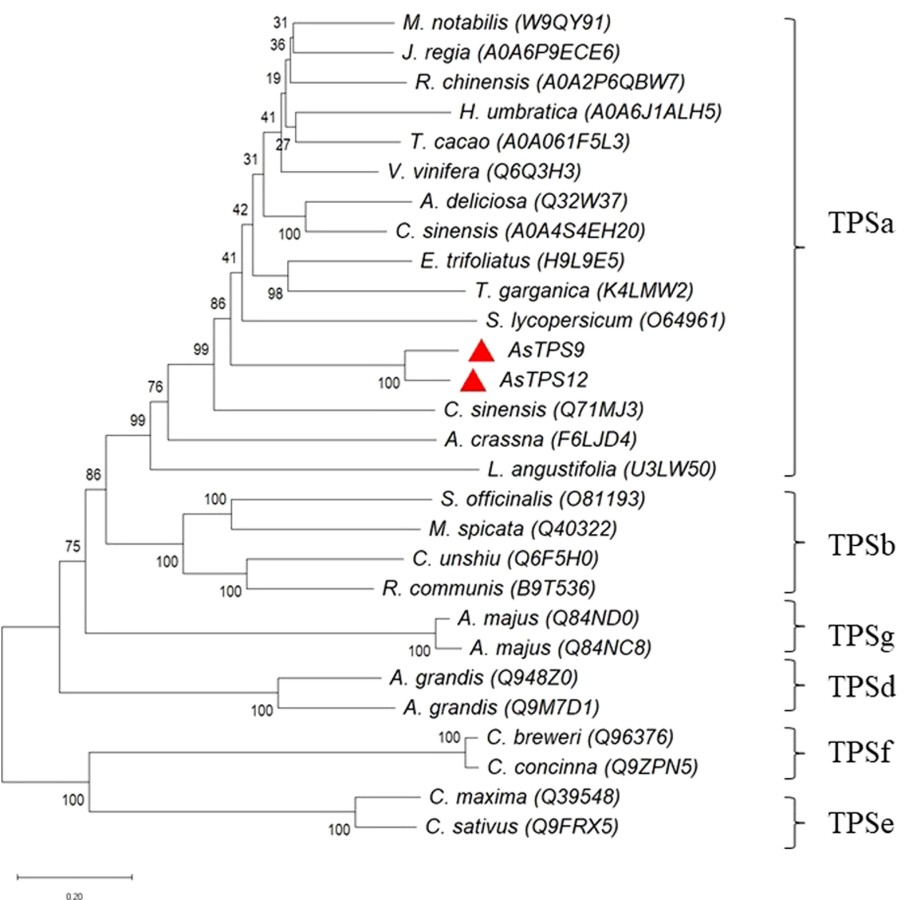

**Figure 2** **Phylogenetic tree illustrating the relationship of the subfamily of TPS family by the neighbor-joining method.** Phylogenetic tree illustrating the relationship of the subfamily of TPS family by the neighbor-joining method. The plant TPSs were from *Herrania umbratica* (*H. umbratica*), *Theobroma cacao* (*T. cacao*), *Juglans regia* (*J. regia*), *Morus notabilis* (*M. notabilis*), *Rosa chinensis* (*R. chinensis*), *Vitis vinifera* (*V. vinifera*), *Actinidia deliciosa* (*A. deliciosa*), *Camellia sinensis* (*C. sinensis*), *Aquilaria crassna* (*A. crassna*), *Lavandulaangustifolis* (*L. angustifolis*), *Salvia officinalis* (*S. officinalis*), *Mentha spicata* (*M. spicata*), *Citrus unshiu* (*C. unshiu*), *Ricinuscommunis* (*R.communis*), *Antirrhinum majus* (*A. majus*), *Abies grandis* (*A. grandis*), *Clarkia breweri* (*C. breweri*), *Clarkia concinna* (*C. concinna*), *Cucurbita maxima* (*C. maxima*), *Cucumis sativus* (*C. sativus*).

able to produce sesquiterpenes. Based on mass spectrum analysis using an equipped database, unknown sesquiterpenes, unidentified sesquiterpene RI 1404 (Rt = 12.642 min), $\beta$-farnesene (Rt = 12.711 min), *cis*-sesquisabinene hydrate (Rt = 15.837 min), and cedrol (Rt = 16.066 min) were detected by GC–MS in TPS9; the products contained major amounts of *cis*-sesquisabinene hydrate (34.61%), $\beta$-farnesene (31.79%), and smaller amounts of unidentified sesquiterpene RI 1404 (19.04%) and cedrol (14.56%). Incubation of TPS12 with FPP resulted in the formation of four sesquiterpenes, $\beta$-farnesene (41.11%), nerolidol (31.07%), $\gamma$-eudesmol (1.77%), and hinesol (26.05%), at retention times of 12.713, 14.842, 16.875, and 17.493 min, respectively (Fig. 4). Among them, nerolidol is a characteristic sesquiterpene present in agarwood which has also been

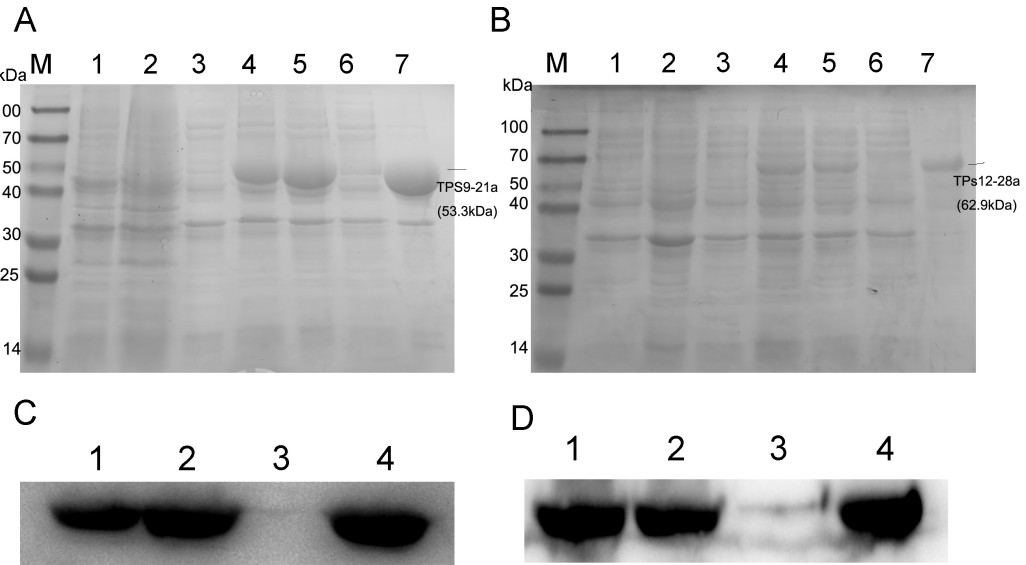

**Figure 3** **SDS-PAGE and Western blot analysis of recombinant enzymes of TPS9, TPS12.** (A) SDS-PAGE analysis of recombinant enzymes of TPS9 in *E. coil* BL21 (DE3). Lane M, protein molecular mass maker (kDa); Lane 1, pET-21a without induction; Lane 2, pET-21a with 0.5 mM IPTG induced for 4 h; Lane 3, TPS9-21a without induction; Lanes 4–5, 0.5 mM IPTG induced TPS9-21a for 2 h, 4 h, respectively. Lane 6, supernatant for TPS9-21a; Lane 7, sediment for TPS9-21a. (B) SDS-PAGE analysis of recombinant enzymes of TPS12 in *E. coil* BL21 (DE3). Lane M, protein molecular mass maker (kDa); Lane 1, pET-28a without induction; Lane 2, pET-28a with 0.5 mM IPTG induced for 4 h; Lane 3, TPS12-28a without induction; Lanes 4–5, 0.5 mM IPTG induced TPS12-28a for 2 h, 4 h, respectively. Lane 6, supernatant for TPS12-28a; Lane 9, sediment for TPS12-28a. (C) Western blot analysis of recombinant enzymes of TPS9 in *E. coil* BL21 (DE3). Lanes 1–2, 0.5 mM IPTG induced TPS9-21a for 2 h, 4 h, respectively. Lanes 3–4, supernatant, sediment for TPS9-21a. (D) Western blot analysis of recombinant enzymes of TPS12 in *E. coil* BL21 (DE3). Lanes 1–2, 0.5 mM IPTG induced TPS12-28a for 2 h, 4 h, respectively. Lanes 3–4, supernatant, sediment for TPS12-28a.

detected in *A. malaccensis* from Cambodia (*Chen et al., 2012*; *Gao et al., 2014*; *Pripdeevech, Khummueng & Park, 2011*). Various studies have found that nerolidol plays a vital role in neuroprotection and has inhibitory effects on leishmanial, schistosomal, malarial, nociceptive, and tumour activities (*Chan et al., 2016*; *De Carvalho et al., 2018*; *Fonseca et al., 2016*; *Iqubal et al., 2019*). In addition, $\gamma$-eudesmol and hinesol have also been found in essential oils from *A. sinensis* (*Chen et al., 2012*; *Gao et al., 2019*; *Xu et al., 2013*) both showing pharmacological anti-tumour effects (*Bomfim et al., 2013*; *Guo et al., 2018*).

Further, the identities of the monoterpene products produced by the sesquiterpene synthases TPS9 and TPS12 were detected by GC-MS, and both incubation of TPS9 and TPS12 with GPP yielded monoterpene products (Table 1 and Fig. 5). Incubation of TPS9 with GPP resulted in the formation of seven monoterpenes, $\beta$-myrcene (2.37%), limonene (1.1%), 3-carene (1.8%), unidentified monoterpene RI 1005 (1.34%), citronellol (39.73%), geraniol (49.73%) and citral (3.93%), with retention times of 5.712, 6.323, 6.436, 6.605, 9.328, 9.739, 9.927 min, respectively. Four monoterpenes, $\beta$-myrcene (3.39%), unidentified monoterpene RI 1006 (2.38%), unidentified monoterpene RI 1089 (2.00%), citral (92.30%), with retention times of 5.724, 6.616, 7.936, 9.769

**Table 1  GC-MS analysis of TPS9 and TPS12 sesquiterpene and monoterpene products.**

| Compounds | $t_x$ | RI | Identification |
|---|---|---|---|
| **TPS9** | | | |
| **sesquiterpene** | | | |
| unidentified sesquiterpene | 12.642 | 1404 | 1505 (*Cavalli et al., 2003*) |
| $\beta$-farnesene | 12.711 | 1408 | 1437 (*Tzakou et al., 2004*) |
| cis-sesquisabinene hydrate | 15.837 | 1547 | 1525 (*Tzakou et al., 2004*) |
| cedrol | 16.066 | 1555 | 1580 (*Basta et al., 2007*) |
| **monoterpene** | | | |
| $\beta$-myrcene | 5.712 | 947 | 955 (*Tepe et al., 2006*) |
| limonene | 6.323 | 987 | 995 (*Tepe et al., 2006*) |
| 3-carene | 6.436 | 995 | 1005 (*Siani et al., 2004*) |
| unidentified monoterpene | 6.605 | 1005 | 944 (*Ghasemi et al., 2005*) |
| citronellol | 9.328 | 1180 | 1217 (*Gkinis et al., 2003*) |
| geraniol | 9.739 | 1206 | 1224 (*Vujisic et al., 2006*) |
| citral | 9.927 | 1217 | 1235 (*Pérez, Navarro & de Lorenzo, 2007*) |
| **TPS12** | | | |
| **sesquiterpene** | | | |
| $\beta$-farnesene | 12.713 | 1408 | 1437 (*Tzakou et al., 2004*) |
| nerolidol | 14.842 | 1512 | 1531 (*Saroglou et al., 2007*) |
| $\gamma$-eudesmol | 16.875 | 1583 | 1621 (*Hammami, Kamoun & Rebai, 2011*) |
| hinesol | 17.493 | 1603 | 1620 (*Li et al., 2006*) |
| **monoterpene** | | | |
| $\beta$-myrcene | 5.724 | 948 | 955 (*Tepe et al., 2006*) |
| unidentified monoterpene | 6.616 | 1006 | 944 (*Ghasemi et al., 2005*) |
| unidentified monoterpene | 7.936 | 1089 | 1005 (*Siani et al., 2004*) |
| citral | 9.769 | 1208 | 1235 (*Pérez, Navarro & de Lorenzo, 2007*) |

min, respectively. The same products of TPS9 and TPS12 were $\beta$-myrcene, and citral. Citronellol and geraniol were the principal monoterpene products of TPS9, but they were not detectable as products of TPS12. Among them, it is reported that myrcene and $\beta$-pinene also the common products generated by various sesquiterpenes synthase with GPP as substrate (*Nagegowda et al., 2008*; *Srivastava et al., 2015*; *Steele et al., 1998*; *Tholl et al., 2005*). Limonene is the principal monoterpene product of both $\delta$-selinene synthase and $\gamma$-humulene synthases in *Abies grandis* (32), and it is also generated by At5g23960 TPS from Arabidopsis. Pinene, myrcene, limonene and cital are detected in the essential oil of Tunisian *Conyza bonariensis* (*Mabrouk et al., 2011*).

## Expression patterns of *TPS9* and *TPS12* in different tissues and growth phases of *A. sinensis*

To investigate the temporal and spatial expression profiles of *TPS9* and *TPS12* in different tissues, the expression levels of *TPS9* and *TPS12* were analysed from transcriptome data in seven tissues, including agarwood, branches, stems, roots, old leaves, young leaves, buds, and flowers (Figs. 6A, 6C). The results showed similar expression profiles for both *TPS9* and *TPS12*, except that *TPS9* was not expressed in stems, young leaves, or buds,

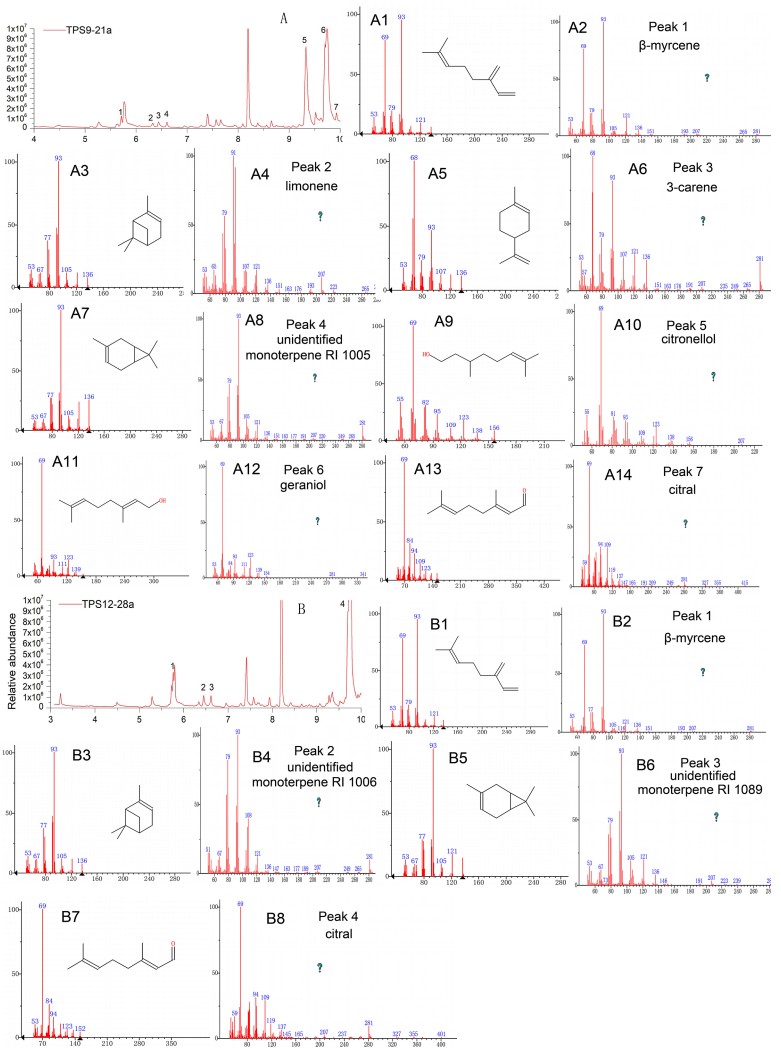

**Figure 4  Gas chromatography-mass spectrometry (GC-MS) analysis of TPS9 and TPS12 sesquiter-pene products.** (A, B) Total ion chromatogram of the products formed by TPS9 and TPS12 with farnesyl diphosphate as a substrate. (A2, A4, A6, A8) Mass spectra of the sesquiterpenes. (A1, A3, A5, A7) authentic standards. (B2, B4, B6, B8) Mass spectra of the sesquiterpenes. (B1, B3, B5, B7) authentic standards.

while *TPS12* was not expressed in branches, stems, or young leaves. The largest amounts of transcripts of *TPS9* and *TPS12* were found in flowers. These results suggest that *TPS9* and *TPS12* probably contribute to flower fragrance, likely playing a role in the synthesis of various aromatic compounds. Furthermore, they seemingly have a function in flower development and other physiological processes occurring in flowers, similar to *At5g44630* and *At5g23960*, which are responsible for the biosynthesis of sesquiterpenes emitted from *Arabidopsis* flowers (*Tholl et al., 2005*). In addition, high expression levels of *TPS9* and *TPS12* were observed in agarwood, which, according to their function, produced volatile sesquiterpenes. These results indicate that the two newly identified sesquiterpene synthases probably have positive roles in agarwood formation.

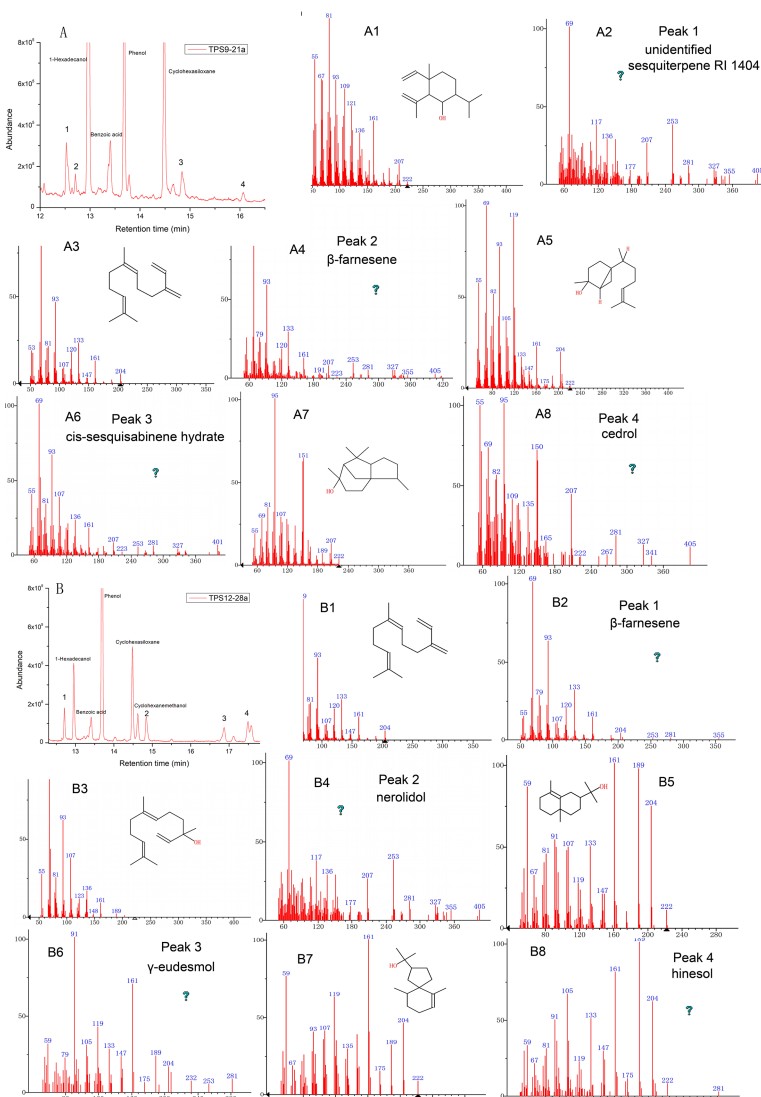

**Figure 5** **Gas chromatography-mass spectrometry (GC-MS) analysis of TPS9 and TPS12 monoterpene products.** (A, B) Total ion chromatogram of the products formed by TPS9 and TPS12 with geranyl pyrophosphate (GPP) as a substrate. (A2, A4, A6, A8, A10, A12, A14) Mass spectra of the monoterpenes. (A1, A3, A5, A7, A9, A11, A13) authentic standards. (B2, B4, B6, B8) Mass spectra of the monoterpenes. (B1, B3, B5, B7) authentic standards.

There is a general consensus that agarwood is only formed in injured *A. sinensis* trees. To further evaluate whether *TPS9* and *TPS12* are involved in agarwood formation, we analysed transcriptome data and whole-tree inducing materials, including different treatment times and layers (healthy layer, H; agarwood layer, AT and A; transition layer, T; and decomposed layer, D) (Fig. 6B). In contrast to *ASS1*, sesquiterpene synthase δ-guanine synthase from *A. sinensis* is immediately and significantly induced by wounding (*Sun et al., 2020*; *Xu et al., 2016*; *Xu et al., 2013*). Both *TPS9* and *TPS12* were induced much later, showing a very low level of expression at 15 days prior to induction. *TPS9* was
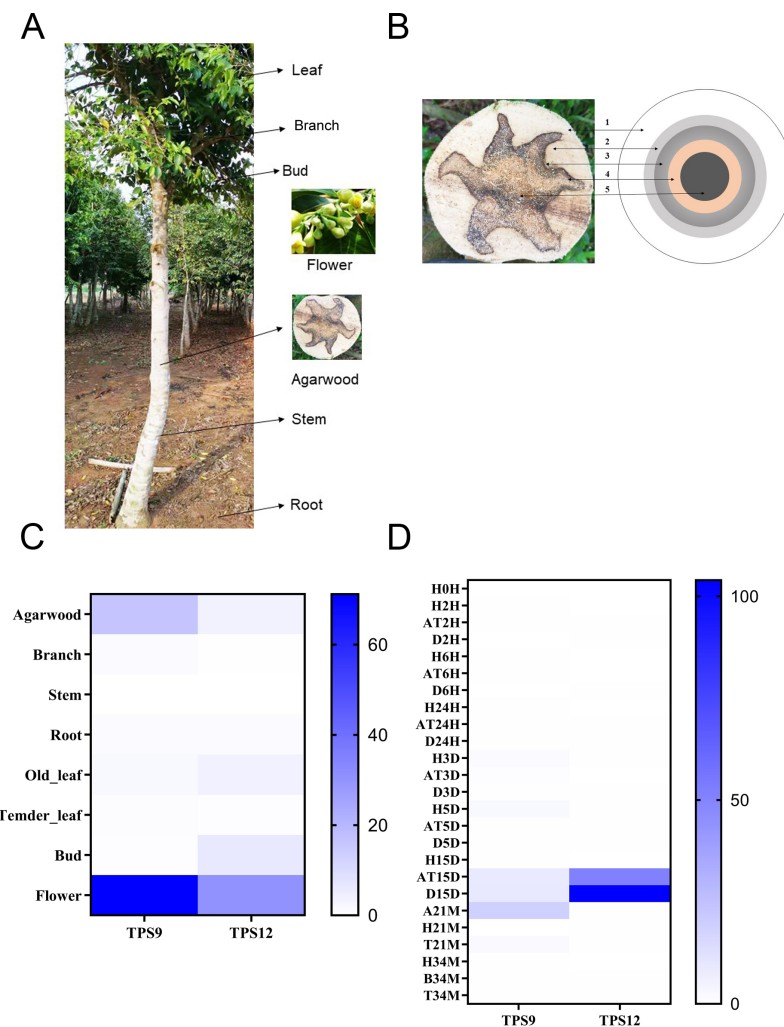

**Figure 6** **Heat map of the *TPS9* and *TPS12* genes expression profiles.** (A). Different tissues of *A. sinensis* trees. (B) Different layers of agarwood in cross section of *A. sinensis* tree. (1) H-the healthy layer; (2) B-the blocked layer; (3) T-the transition layer; (4) AT-the agarwood layer; (5) D-the decayed layer after Agar-Wit treatment. (C) Expression patterns of *TPS9* and *TPS12* in different tissues. (D) Expression patterns of *TPS9* and *TPS12* in different times under Agar-Wit treatment and different layers. All gene expression levels were transformed to scores ranging from 0 to 100 and were colored different shades of blue to represent low, moderate, or high expression levels, respectively.

expressed in the agarwood and the decomposed layers 15 days after wounding treatment, and lasted until 21 month, with the highest expression level at 21 month (Fig. 6D). The relative expression level of *TPS12* was extremely high in agarwood and decomposed layers at 15 days, and the expression was at a low level at later stages.

## The response of *TPS9* and *TPS12* to various abiotic stress conditions

To examine whether *TPS9* and *TPS12* would respond to various abiotic stress, we analysed well-grown *A. sinensis* calli treated with heat, 100 μM MeJA, 100 μM SA, 100 μM ABA, 400 mM mannitol, 50 mM $H_2O_2$ and 300 mM NaCl at various times, and then

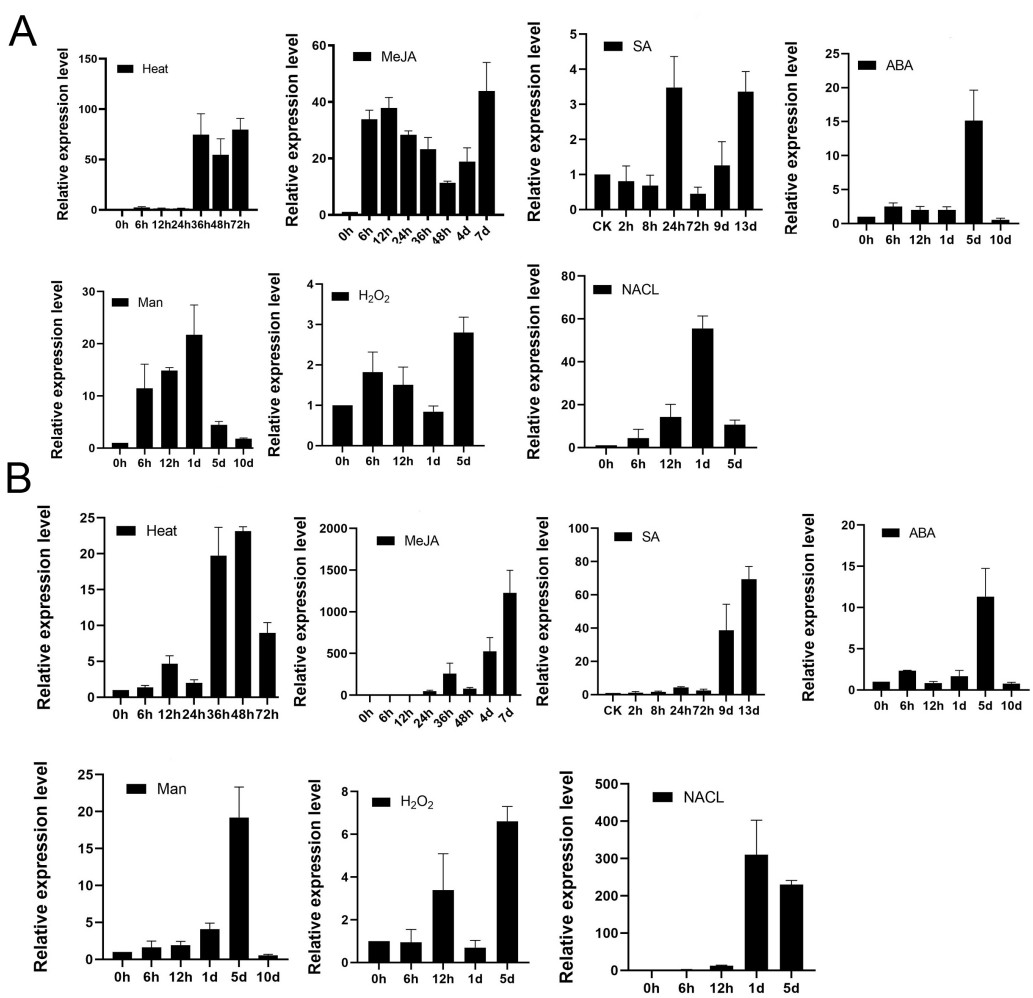

**Figure 7** **The expression analysis of *TPS9* and *TPS12* in *A. sinensis* calli at various time points under various stresses treatment.** (A) The expression of *TPS9* in response to various additional stresses. (B) The expression of *TPS12* in response to various additional stresses. GADPH was used as a reference gene. Heat, transform 25 °C to 42 °C; MeJA (methyl jasmonate), 100 μM treatment; SA (salicylic acid), 100 μM treatment; ABA (abscisic acid), 100 μM treatment; Man (mannitol), 400 mM treatment; H₂O₂, 50 mM treatment; NACL, 300 mM treatment. Three independent biological replicates were performed.

used qRT-PCR to study the resulting transcript levels of *TPS9* and *TPS12*. As shown in Fig. 7, *TPS9* and *TPS12* were induced by all additional stresses, and they showed different expression under the stresses. TPS9 seems to be more sensitive to heat and MeJA treatment while TPS12 induces dramatically after MeJA and NaCl treatment.

We previously reported that treatment with MeJA, heat shock, or salicylic acid (SA) induced the expression of sesquiterpene synthase and the production of sesquiterpenes (*Liao et al., 2015a*; *Liao et al., 2015b*; *Liu et al., 2015b*), and MeJA, SA are also known as universal inducer of plant defensive metabolite production, they are response to induce genes expression to biosynthesis secondary metabolites in plants (*Hassanein, 2010*; *Schenk et al., 2000*). Both *TPS9* and *TPS12* were positively induced by all three treatments but

*TPS12* was more sensitive to MeJA than *TPS9*, and significant induction (approximately 60-fold) of *TPS12* was observed in response to stress at 24 h after treatment, after which there was a remarkable increase (nearly 1,000-fold) at 7 d after treatment, with the trend still continuing. Meanwhile, the expression level of *TPS9* transcripts increased by approximately 40-fold at 6 h after treatment and the induced transcription level was maintained as the treatment continued. As for heat treatment, it increased the transcript levels of both *TPS9* and *TPS12* but the changes were not as significant as those observed under MeJA treatment. The expression of *TPS9* and *TPS12* both increased first, until levels peaked at 80- and 20-fold, respectively, at 36 h after treatment, and then decreased. Under salicylic acid (SA) treatment, the relative expression of *TPS9* showed an up-down-up pattern and peaked at 24 h with an expression level approximately four times higher than that recorded prior to treatment. Meanwhile, the expression level of *TPS12* remained stable during 72 h, increasing significantly, approximately five-fold, after nine days; then, the expression increased to a maximum level, approximately 80-fold greater than that observed for controls, at 13 days after treatment. ABA plays a central role in responses to biotic and abiotic stresses (*Smet et al., 2006*). Under ABA treatment, *TPS9* expression level was increased to about 3-fold immediately and maintained stable until day 1, peaked at about 15-fold at day 5 and rapidly decreased to minimum value at day 10, which in accord with *TPS12* expression. Mannitol is suitable for screening crop contained tolerance drought stress (*Jing et al., 2009*) and it has been found to be effective as a water stress-inducing agent in various plants, including *Pelargonium* (*Hassanein, 2010*), cassava (*Jolayemi, Opabode & Badara, 2018*), soybean (*Neto et al., 2004*). The *TPS9* expression increased till maximum about 20-fold at day 1 and then the expression was gradually decreased until day 10. On the other hand, *TPS12* gradually increased and peaked at day 5, and then rapidly decreased until day 10. For $H_2O_2$ treatment, *TPS9* transcript level peaked at day 5 for about 3-folds, and *TPS12* expression showed up-down-up pattern and peaked at day 5. For NaCl treatment, the expression of *TPS9* and *TPS12* showed similar pattern, they both gradually increased and peaked at day 1, and then decreased at day 5.

These results suggest that *TPS9* and *TPS12* are induced at all treated stresses, implying they might play a crucial role in protecting plants from environment abiotic stresses. Furthermore, under all stresses treatment, TPS9 and TPS12 were typical genes that are induced in the later stages of the plant response to stress conditions.

## DISCUSSION

As agarwood formation occurs occasionally and takes a long time in natural environments, and given that the demand for agarwood is continuously on the rise, naturally, at present, such demand already exceeds market supply. In response to this situation, various artificial methods have been tested to induce agarwood formation, including cutting (*Li et al., 2016*), chemical wounding (*Zhang et al., 2013*), and fungal inoculation (*Gibson, 1977*), all of which are widely used in South and Southeast Asia. However, artificial induction produces products with low yield and quality. Therefore, determining the mechanism of agarwood formation and finding new ways to improve the yield and quality of artificial agarwood are current research priorities.

Previous studies have reported that the *BdTPS* gene is expressed in various tissues of *Brachypodium distachyon*, including roots, stems, leaves, and spikes (*Wang, Ouyang & Wang, 2019*). In our study, *TPS9* and *TPS12* were mainly expressed in the flowers and agarwood which are the two major organs that emit floral volatiles, suggesting that *TPS9* and *TPS12* probably contribute to fragrance production by synthesising sesquiterpenes. In Arabidopsis, two TPS genes show similar functions, which are responsible for the biosynthesis of virtually all detectable sesquiterpenes emitted by flowers (*Tholl et al., 2005*). Combined with their expression in the different layers of the Agar-Wit treatment, both *TPS9* and *TPS12* were induced upon stress treatments, although long after treatment application, contrary to *ASS1*, which is a typical inducible gene and upregulated almost 200-fold over the normal level after only 1.5 h of mechanical wounding (*Xu et al., 2021*). *TPS9* and *TPS12* were found mainly expressed in the agarwood layer after 15 days. These three sesquiterpene synthases, *TPS9*, *TPS12,* and *ASS1*, might play different roles in agarwood production at different stages. Thus, *ASS1* might be responsible for agarwood formation, while *TPS9* and *TPS12* are probably involved in agarwood accumulation.

Numerous studies have revealed that *TPS* genes play vital roles in protecting plants from various abiotic stress conditions (*Garg et al., 2002*; *Ge et al., 2008*). Consistently, *Avonce et al. (2004)* reported that drought tolerance of *Arabidopsis* plants overexpressing *AtTPS1* was significantly improved, while overexpression of *OsTPS1* increased rice tolerance to abiotic stress (*Ge et al., 2008*). In this study, results of qRT-PCR showed that *TPS9* and *TPS12* were both positively induced by MeJA, heat stress, SA, ABA, mannitol, $H_2O_2$ and NaCl (Fig. 7). *TPS9* and *TPS12* expression showed a more significant increase in response to MeJA treatment and NaCl treatment, and they all increased sharply after 24 h, which supports the hypothesis that these genes are induced at later stages during the plant response to stress. Conversely, as for *ASS1*, an *A. sinensis* gene whose upregulation occurs very soon after stress treatments, MeJA significantly promoted its upregulation, whereby its expression at 6 h was approximately 180 times higher, peaking at 12 h at a level approximately 600 times higher (*Sun et al., 2020*), implying that *ASS1* responds rapidly and markedly to stress treatments. Apparently, these three TPSs, TPS9, TPS12, and ASS1, play different roles in protecting plants upon stress treatments.

In the experiments reported herein, Kovats retention index showed some differences between the test values and the literature values. Previous studies have reported the retention time of the compounds was affected by the column types and carrier gases (*Goodner, 2008*; *Jouyban et al., 2011*). The chromatographic column used in the formulation of RI is a HP-5MS column, while DB-5MS column was used in our experiments. TPS9 was able to liberate *cis*-sesquisabinene hydrate (34.61%) and *β*-farnesene (31.79%) as major products, and smaller amounts of unidentified sesquiterpene RI 1404 (19.04%) and cedrol (14.56%). *Goodner (2008)* has reported that 3% and 5% were chosen as reasonable limits based on variations of reported retention indices. Hence, the literature value exceed ca. 5% have been identified as "unidentified compound". Among them, *cis*-sesquisabinene has not been previously detected in *A. sinensis*. Although cedrol has been found in the essential oil from *A. sinensis* and *A. crassna* (*Gao et al., 2019*; *Thanh et al., 2015*), and can be formed by sesquiterpene synthases isolated from other plant

species, such as *Artemisia apiacea* (*Kanagarajan et al., 2012*) and *Artemisia maritima* (*Muangphrom et al., 2018*) yet, there are no previous reports of a sesquiterpene synthase in *A. sinensis* that can catalyze cedrol.

Meanwhile, TPS12 produced $\beta$-farnesene (41.11%), nerolidol (31.07%), hinesol (26.05%), and traces of $\gamma$-eudesmol (1.77%), among which nerolidol is a characteristic sesquiterpene from agarwood (*Chen et al., 2012*; *Rohmer, 1999*). Nerolidol and $\beta$-farnesene can also be catalysed by *As-SesTPS* from *A. sinensis* (*Ye et al., 2018*). In turn, $\gamma$-eudesmol and hinesol have been found in essential oils from *A. sinensis* and *A. crassna* (*Thanh et al., 2015*; *Xu et al., 2013*), but no studies have verified whether sesquiterpene synthases in *Aquilaria* Lam. can catalyze their formation. Various studies have reported that nerolidol plays a vital role in neuroprotection and has a positive effect on the inhibition of leishmanial, schistosomal, malarial, nociceptive, and tumour activities (*Chan et al., 2016*; *De Carvalho et al., 2018*; *Fonseca et al., 2016*; *Iqubal et al., 2019*). $\beta$-Farnesene is the only common reaction product of TPS9 and TPS12 that is produced in large quantities; furthermore, it was not found in agarwood but in some flowers from other plant species (*Mabrouk et al., 2011*; *Yu et al., 2011*), which is consistent with our observation of the high expression levels of *TPS9* and *TPS12* detected in flowers. Thus, three sesquiterpene synthases were isolated from *A. sinensis* (TPS9, TPS12, and As-SesTPS), all belonging to the TPSa subfamily. Their products vary to a large extent with only one common product, $\beta$-farnesene (*Ye et al., 2018*), implying a diversity of enzymatic functions for the three enzymes.

As shown in Table 1 and Fig. 5, TPS9 and TPS12 were able to generate monoterpene products with GPP as substrate in the contrary to the sesquiterpene synthases Catps1, LaCADS, and ReSeTPS10As, which failed to produce an product upon GPP induction despite containing similar sequences as TPS9 and TPS12. (*Jullien et al., 2014*; *Sharon-Asa et al., 2003*; *Xie, Kirby & Keasling, 2012*). Furthermore, LaCADS also failed to convert geranylgeranyl pyrophosphate (GGPP) into diterpenes (*Jullien et al., 2014*). Generally, all monoterpene synthases were thought to be localized to plastids to be able to use GPP as substrate, while sesquiterpene synthases were thought to be localized to the cytosol where they use FPP as their substrate and the use of GPP by sesquiterpene synthases only occurred *in vitro* (*Chen et al., 2011a*). Therefore, the formation of these compounds by the enzyme *in vivo* is rather unlikely, as the TPS9 and TPS12 protein lacks a transit peptide and is therefore not expected to be present in plastids, where GPP is thought to be produced.

## CONCLUSIONS

In this study, two novel genes, *TPS9* and *TPS12*, which encode sesquiterpene synthases, were amplified from the cDNA of *A. sinensis* calli. *TPS9* and *TPS12* were both expressed mainly in flowers, and reached the highest expression level in the agarwood layer at 15 days after wounding, suggesting that they are typical genes induced at the later stages when plants respond to stress, which accord with the expression patterns of TPS9 and TPS12 after treated with MeJA, SA, ABA, mannitol, H2O2 and NACL. Both TPS9 and

TPS12 efficiently converted FPP and GPP to particular products. TPS9 catalyses the conversion of FPP to $\beta$-farnesene, *cis*-sesquisabinene hydrate, cedrol and unidentified sesquiterpene RI 1404. Meanwhile, four sesquiterpenes were found after incubation of TPS12 with FPP, including $\beta$-farnesene, nerolidol, $\gamma$-eudesmol, and hinesol. Seven monoterpenes were detected when TPS9 was incubated with GPP, including $\beta$-myrcene, limonene, 3-carene, citronellol, geraniol, citral and unidentified monoterpene RI 1005. On the other hand, TPS12 converted GPP into four monoterpenes, $\beta$-myrcenecitral, unidentified monoterpene RI 1089 and unidentified monoterpene RI 1006. Further studies should be conducted to establish transgenic systems to explore the physiological functions of TPS9 and TPS12 and their regulatory roles in agarwood formation. This study lays a solid theoretical and experimental foundation for future research on gene function and provides clues for sesquiterpene biosynthesis and agarwood accumulation.

### Funding

This research was financially supported by the National Science Foundation of Beijing (7222286), the Key Research and Development Program of China (2018YFC1706400), the National Natural Science Foundation of China (82073967, 82173925). The funders had no role in study design, data collection and analysis, decision to publish, or preparation of the manuscript.

### Grant Disclosures

The following grant information was disclosed by the authors:
National Science Foundation of Beijing: 7222286.
Key Research and Development Program of China: 2018YFC1706400.
National Natural Science Foundation of China: 82073967, 82173925.

### Competing Interests

The authors declare there are no competing interests.

### Author Contributions

- Cuicui Yu performed the experiments, analyzed the data, prepared figures and/or tables, authored or reviewed drafts of the article, and approved the final draft.
- Shixi Gao performed the experiments, prepared figures and/or tables, and approved the final draft.
- Mei Rong analyzed the data, prepared figures and/or tables, and approved the final draft.
- Mengjun Xiao analyzed the data, prepared figures and/or tables, and approved the final draft.
- Yanhong Xu conceived and designed the experiments, authored or reviewed drafts of the article, and approved the final draft.

● Jianhe Wei conceived and designed the experiments, authored or reviewed drafts of the article, and approved the final draft.

## DNA Deposition

The following information was supplied regarding the deposition of DNA sequences:

The TPS9 and TPS12 sequences are available at GenBank: MZ969011, MZ969013, respectively.

## Data Availability

The raw measurements are available in the Supplemental Files.

## Supplemental Information

Supplemental information for this article can be found online at http://dx.doi.org/10.7717/peerj.15818#supplemental-information.

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
