# Peer review of "Identification and characterization of novel sesquiterpene synthases TPS9 and TPS12 from Aquilaria sinensis"

_PeerJ, doi:10.7717/peerj.15818_

## Round 0.1 · original submission · Major Revisions

Three reviewers evaluated your manuscript and, in general, support its publication. However, you need to experimentally confirm the identification of metabolites.

In addition, please correct language errors.

Reviewer 1 ·

Basic reporting

In this manuscript, the authors described cloned genes of two two terpene synthases from Aquilaria sinensis and tested the in-vitro activity of the enzymes using two different substrates. The product spectrum was analyzed by GC-MS. They also checked the expression of the corresponding genes in different parts of the plant.
In general, this manuscript is understandable. However, several grammar mistakes should be corrected. The manuscript would profit from further language editing. The introduction shows the context correctly and the manuscript is structured well. Literature appears well referenced. I found a few aspects that could be improved on:
Here are some suggestions to improve figures:
• Figure 1. Increase resolution
• Figure 2: Put species names in italics
• Figure 4: Figure has unacceptable resolution in the final PDF, could be a problem of PDF generation.
• Figure 5: Figure has unacceptable resolution in the final PDF.
• Figure 6: I would suggest to show panels C and D first (as A and B). Then, the legend will become obvious for the reader more quickly.

• Supplement_figure_1: increase resolution

Raw data supplied (see PeerJ policy).
• Supplement_figure_2: This figure is meaningless for me. To understand motifs, the name of the motif should be given as well as show coordinates of each protein. Please also add a short description of methodology used.
• Supplemental Table 1: Please add a short description of the methodology used.
• MS data: it would be better to also provide MS-data in an open, text-based format instead of just screen shots of spectra that are already shown in the figure.
• Gene accession numbers work

Experimental design

Overall, the experimental design is ok and methods are described well. However, the identification of metabolites just via database matches of mass spectra is doubtful.
The matches of mass spectra of terpenes to the NIST database are not sufficient to make reliable identifications. Ideally, identifications should be confirmed using synthetic standards. At least, the Kovats retention index should be used to double-check compound annotations. This issue urgently needs to be addressed before publication.

Validity of the findings

Generally, the manuscript is quite descriptive and lacks mechanistic insights. However, it might be interesting for a specialized audience and set to ground for further mechanistic studies into the ecological function of the terpenes produced.

Additional comments

• Title: I would also take out the strain designation “(Lour.) Gilg (#72847)” from the title and add it to the abstract instead.
• Line 27 of abstract: delete “is clear”.
• Line 29: change “detected” to “examined”
• Line 34: change “incubated” to “incubation”
• Line 39: change “corresponding” to “response”
• Lin 62: Rephrase to remove “is better known than that of chromone derivatives”
• Line 66: change “able to” to “are able to”
• Line 80: Maybe name these sesquiterpene synthases?
• Line 87: “Various stresses treatment” Better: “various stress treatments”  check hole manuscript
• Line 90: Change “clone” to “heterologously produced”
• Line 91/99: the authors should mention what “Agar-Wit treatment” actually means already in the introduction
• Lines 170 to 179: Please, create a separate section like “Heterologous protein production and enzymatic assays”.
• Line 245: “four new sesquiterpenes”. Maybe say: “unknown sesqiterpenes”
• Line 285: “further, floral scents played a vital role in attracting insects and other animal pollinators.” This statement does not really fit to the flow of thoughts here. Please shift or delete it.
• Line 393: Change “catalysed” to “formed”.
• Line 402: “can catalyse them”. Better: “can catalyze their formation”. Check whole manuscript.
• Line 431: “when plant response” to "when plants respond”

·

Basic reporting

The article was reporting on the successfully identified and characterized of novel sesquiterpene synthase TSP9 and TSP 12. Although the study has been conducted in standard manner however some modification and revise should be made as mentioned in the pdf file.

Experimental design

Please refer to the comment in pdf file

Validity of the findings

Please refer to pdf file

Additional comments

Please refer to pdf file

·

Basic reporting

Agarwood is popular globally due to its high commercial value as an incense, as a scent, as an ornamental wood, and as a traditional medicine. However, agarwood formation occurs slowly under external stimuli. Previous studies have demonstrated that sesquiterpenoids and chromones are the main active constituents in agarwood. In this study, the authors characterized two terpene synthases TPS9 and TPS12. The expression patterns of TPS9 and TPS12 in different tissues were also investigated by transcriptomics and the data showed that TPS9 and TPS12 showed higher expression in flowers and agarwood.

Experimental design

Overall, the reviewer reads this manuscript well. The conclusions are well supported by the designed experiments and the corresponding data observed.

Validity of the findings

The authors biochemically characterized two terpene synthases TPS9 and TPS12 from agarwood. The expression pattern of these two TPS were also reported via transcriptomics. Furthermore, the regulation of these two TPS under different external stimulus were investigated.

Additional comments

There are several minor errors that the authors are suggested to revise to improve the quality of this manuscript.
1. Page 6 Line 52, “Consequently, and to protect plant genetic resources” the “and” should be removed.
2. Page 10 Line 226 “TPS12-encoded protein” TPS12 should be italicized, a gene encodes a protein.
3. In the whole manuscript Rt=12.642 there should be blanks before and after =.
4. The authors should check the whole manuscript, such epi-, cis-, should be italicized.

---

## Round 0.2 · Minor Revisions

Please address the comments of the reviewer.

In addition, please provide evidence for professional proofreading before resubmission.

·

Basic reporting

The authors have addressed most of the issues raised in the last peer review. The quality of this manuscript has been improved. However, there are still a batch of obvious typos or mistakes that need to be fixed before the publication.

Experimental design

The experiments are well designed to examine the functions of two new terpene synthase genes.

Validity of the findings

The conlusions are generally well supported by the designed experiments and the subsequent data.

Additional comments

1. The name of the first author in Pages 4 and 5 is different.
2. Page 5 line 28, the sentence “Although the sesquiterpene biosynthesis pathway and sesquiterpene synthases (TPSs) are well-known key enzymes of the pathway” is confusing. Biosynthesis pathway are enzymes? It is a bad sentence need to be revised, for example “Although sesquiterpenes are well-known to be biosynthesized by sesquiterpene synthases (TPSs), to date…”
3. Page 5 line 38-39 It is a bad sentence. “Further, RT-PCR analysis showed they are typical genes mainly expressed during later stages of stress response is better known than that of chromone derivatives”. Which can be considered to revise as “Further, RT-PCR analysis showed TPS9 and TPS12 are typical genes mainly expressed during later stages of stress response, which is better known… ”

---

## Round 0.3 · Major Revisions

Please provide your mass spectra in an adequate resolution and, ideally, also deposit them in a public repository such as zenodo.org (free of charge).
The identification criteria of compounds must be clearly explained and justified. Doubtful identifications should be avoided.
Please provide high-resolution graphics that are suitable for publication and proofread your text for eliminating grammar errors.

Reviewer 1 ·

Basic reporting

In general, this manuscript has improved significantly after my last review. However, some of the points I criticized previously have not been addressed. Also the identification of compounds needs major revisions (see validity of findings).
Here are some suggestions to improve the manuscript:
• Figure 1. Increase resolution
• Figure 4 and 5: Both figures are of unacceptably low resolution. Please increase the size and resolution of the mass spectra. This is critical for proper identification of the compounds.
• Figure 6: The panels were flipped according to my suggestion, but the legend was not updated accordingly. Please adjust the legend (panels A,B,C,D).
• Figure 7: Legend in panel TPS9 is cut

Raw data supplied (see PeerJ policy).
• Supplement_figure_2: This figure is still meaningless for me. To understand motifs, the name of the motif should be given as well as show coordinates of each protein. Please also add a short description of methodology used.

Further suggestions:
• Abstract: Eliminate space character after “derivatives .”
• Line 80: “et al.”: Check journal style if Italics should be used or not and format uniformly.
• Line 353: suit -> suitable
• Line 354: have been found -> has been found

Experimental design

Concerning my previous criticism of compound identifications, I appreciate that Kovats retention indices have been determined for all metabolites.

Validity of the findings

Some identifications of compounds are still doubtful since Kovats retention indices deviate significantly from the literature for some compounds. More specifically, the deviation from the literature value should not exceed ca. 1%. In these cases, I would suggest eliminating the identification (along with the doubtful compound name) and call the compounds in question something like “unidentified sesquiterpene RI 1404”. The similarity of MS spectra to the one of a known compound found in the database could still be mentioned and discussed, but I would strictly avoid doubtful identifications. A missing identification is not necessarily a weakness of this study, but erroneous identifications would be problematic. When identifications remain open, future studies can build on this paper and elucidate the structure of the unidentified compounds further.

Furthermore, because of the poor resolution of figures 4 and 5, comparing mass spectra to the reference is not possible, thus, the quality of identifications cannot be adequately assessed. Therefore, the resolution of the mass spectra must be improved before publication.

·

Basic reporting

The authors have addressed the issues properly.

Experimental design

no comment

Validity of the findings

no comment

---

## Round 0.4 · accepted · Accept

Thank you very much for your excellent contribution!

Reviewer 1 ·

Basic reporting

The authors have addressed all of my concerns. The paper can now be published.

Please correct the following typos:

Line 32: delete "a" in front of "multiproduct enzymes"
Line 414: change: exceed --> exceeding

Experimental design

No change required

Validity of the findings

All conclusions are now justified